# Feeding growing button mushrooms: The role of substrate mycelium to feed the first two flushes

Anton S. M. Sonnenberg[1]*, Johan J. P. Baars[1], Gerben Straatsma[2], Patrick M. Hendrickx[1], Ed Hendrix[1], Chris Blok[3], Arend van Peer[1]

1 Plant Breeding, Wageningen University and Research, Wageningen, The Netherlands, 2 Department of Environmental Sciences, Wageningen University and Research, Wageningen, The Netherlands, 3 Greenhouse Horticulture & Flower Bulbs, Wageningen University and Research, Bleiswijk, The Netherlands

☙ These authors contributed equally to this work.

* anton.sonnenberg@wur.nl

**Data Availability Statement:** All raw data are available in a public repository accessable at DOI 10.4121/19285106.

## Abstract

A number of experiments were done to further our understanding of the substrate utilization in button mushroom crops (*Agaricus bisporus*). An analysis of the degradation of dry matter of the substrate during a crop cycle revealed that for pin formation the upper 1/3rd layer is used, for the production of flush one all layers are involved and for flush two mainly the lower 1/3 layer is used. A reduction in substrate depth leads to a decrease in yield/m² but an apparent increase in yield per tonne of substrate with a lower mushroom quality. A short daily interruption of the connection between the casing soil with the substrate results in a delay of the first flush. Interruptions with only part of the substrate did not lead to delay in production. Daily interruption of the connection with all or only part of the substrate leads to a shift in yield from flush one to flush two but the total yield remains unchanged. The mycelial biomass in the substrate increases from filling up to pinning, has a steeper increase during flush one, and is levelling off during flush two, indicating that in the period of venting and up to/including flush one, enzymes are secreted by growing hyphae generating nutrients to feed a fixed amount of mushroom biomass for two flushes. A sidewise extension of the substrate (without casing soil, thus not producing mushrooms) showed that the substrate at a distance more than somewhere between 20–50 cm away from the casing soil does not contribute to feeding mushrooms in the first two flushes. The observations are discussed with respect to relevant previous research.

## Introduction

Substrate (compost) plays a prominent role in the cultivation of button mushrooms (*Agaricus bisporus*) since it is one of the main determinant factors for yield and quality of mushrooms. In the majority of Western countries button mushroom substrate is prepared from horse and/ or chicken manure, wheat straw and gypsum (basic mixture). A wetted mixture of the raw

**Funding:** Grant numbers PT-14444, PT-13639, PT-14831 Authors receiving all grants" AS, JB & CB Horticultural board; https://www.productschaptuinbouw.nl/ The funders had no role in study design, data collection and analysis, decision to publish, or preparation of the manuscript.

**Competing interests:** The authors have declared that no competing interests exist.

materials is fermented indoors (tunnels) in two phases which leads to a selective medium suitable for the vegetative growth of *A. bisporus* [1]. The fermented substrate (phase II) is subsequently inoculated with spawn and fully colonized during 14–16 days in tunnels at 24–26˚C. This spawn-run substrate (phase III substrate) is used by growers to fill shelves in growing houses where it is covered with a nutrient poor casing layer consisting of peat and lime. This layer is needed to induce mycelium to form fruiting initials (pins) after venting the rooms by lowering the air temperature and $CO_2$ concentration [2]. Most growers produce only two flushes which corresponds with a reduction in phase III substrate of dry matter by 16%, equivalent to 22% of the organic matter [3]. Phase III substrate, of which thus only a small part is used, represents ca 25% of the production costs of fresh hand-picked mushrooms [PERSONAL COMMUNICATION, JOS HILKENS, ADVISIE]. Transport costs play a significant role since almost as much phase III substrate is transported to, as is retrieved as spent substrate from growers, which makes this cultivation system not an example of an efficient (and sustainable) production system. In addition, the disposal of spent mushroom substrate can be problematic [4] and is a cost factor [AdVisie]. The main reason to use this system is the availability of relatively cheap raw materials that generate, as mentioned before, after fermentation a selective medium for the cultivation of button mushrooms.

A considerable number of research papers on the preparation and utilisation of button mushroom substrate have been published, from basic research to advanced research, including omics [3, 5–16]. This has increased our knowledge on the fermentation process and on transport of nutrients to mushrooms. It has, however, not resulted in a significant change in the production system itself. Substrate is still made in a similar way as has been done for the last 50 years. Our knowledge on how exactly the cultivation system works is still limited and it is not clear what might be opportunities for improvement or alternatives for the present system, especially the substrate.

To further our understanding on how the different layers are utilized, we examined in a first experiment the degradation of the substrate starting from filling phase III substrate until after flush two, at three different depths in the substrate. We measured, in addition, the ergosterol content of the substrate during two flushes as an estimate of changes in mycelial biomass during the crop cycle. We also interfered with the system in three different ways. In the second experiment we varied the amount of substrate/m$^2$ using the common and reduced depths of phase III substrate. In the third experiment, we shortly interrupted the contact of the casing soil (developing mushrooms) with all or part of the substrate. This was done by using iron grids positioned between the casing soil and substrate or a grid positioned at one third or two third depth of the substrate. The interruptions were done daily by shortly lifting the grids to break mycelium contacts between layers, starting at venting up to pin formation (five consecutive days) or from venting up to the first flush (ten consecutive days). In the fourth experiments we added extra substrate by using trays with varying lengths and heights, but a fixed area covered with casing soil (area of cultivation). In this way we varied the amount of substrate permanently available (different filling depth or different sidewise extension of substrate) or temporarily available (by interruptions of contact of developing mushrooms with all or part of the substrate). The effects were measured on number, piece weight and dry weight of mushrooms and the yield per flush. In addition to yield, the quality of the harvested products is an important economic factor for growers. Previous research has shown that ripening/maturation of mushrooms is triggered by competition for nutrients [17]. Straatsma et al. monitored the formation of mushroom biomass in time by removing trays at different time points and picking the complete standing crop of a tray until growth stops. A semi-logarithmic plot showed an exponential growth of biomass until 200 gram/kg substrate. After that point growth becomes linear indicating a competition for nutrients between developing mushrooms. The

start of this linear increase in biomass is characterized by ripening/maturation of mushrooms, i.e., a stretching and subsequent torning of the velum and finally opening of the mushroom caps. A maximum of biomass is reached at 400 gram mushrooms/kg substrate. It is thus relevant to see what influence the experiments with a permanent and timely varied availability of substrate (nutrients) have on quality.

The observations are discussed in how far these contribute to our understanding of the system and how our observations relate to previous research.

## Materials & methods

### Cultivation experiments

In all experiment phase III substrate and casing soil from CNC (Milsbeek, the Netherlands) was used. The CNC substrate formulation has been described previously [1]. Substrate was colonized with the *A. bisporus* variety A15 and supplemented with 15 kg/tonne McSubstradd (Havens, Maashees, the Netherlands). Experiment 1 and 2 were carried out with different batches of substrate and in different growing rooms. Experiment 3 and 4 were done with the same batch of substrate in the same growing room. The climate in the cultivation rooms is quite homogeneous and trays were placed in a randomized block design. Mushrooms were harvested during two flushes and divided in quality classes common for the Dutch market: Class I fine (closed caps with diameter 20–40 mm), Class I middle (closed cap with diameter 40–64 mm) and Class II (velum stretched/sometimes open). The experimental setups are shown in Table 1 and statistics were carried out using SPSS 28 (IBM) and details shown in S1 Appendix.

### Experiment 1: Substrate degradation at different depths

Trays with a surface area of 0.2 m$^2$ (inside dimensions 0.56 x 0.36 cm) were filled with 16.5 kg phase III substrate (82.5kg/m$^2$). At filling, iron grids were placed between the casing soil and the substrate or at one of two depths in the substrate dividing the layers in three equal parts of 5.5 kg substrate. The grids used in all experiments had a square mesh width of 3.6 cm and a wire thickness of 3 mm. The substrate of each tray was covered with 5 cm casing soil which was mixed with 100 gram phase III substrate. In order to maintain moisture content of the casing soil optimal, the crop of Experiment 1 and 2 was watered at day 1 (filling), and day 2 and 3 with 2 L/m$^2$. Same watering was done after the first flush (day 16 and 17). Trays were sacrificed for analysis at venting, start of pin formation, middle of flush one, end of flush one, in-between-flush one and two, middle of flush two, and end of flush two (Fig 1). For each time point two trays were analysed. The cultivation of two flushes was carried out as described previously [18]. At each time point indicated in Fig 1, the three layers of substrate were weighed

**Table 1. Experimental setup of the four experiments described in M&M.**

| Experiment 1 | | | Experiment 2 | | | Experiment 3 | | | Experiment 4 | | |
|---|---|---|---|---|---|---|---|---|---|---|---|
| Factor | Level | N | Factor | Level | N | Factor | Level | N | Factor | Level | N |
| **Grid depth** | 0 | 2 | **Substrate (kg/m$^2$)** | 52 | 6 | **Grid depth** | 0 | 4+8 | **kg substrate underneat casing per tray** | 6.7 | 4 |
| | 1 | 2 | | 67 | 6 | | 1 | 8 | | 9.6 | 8 |
| | 2 | 2 | | 82 | 6 | | 2 | 8 | | 14.4 | 4 |
| | | | **N-supplement (kg/ton substrate)** | 20 | 6 | **Distruption-length (days)** | 0 | 4 | | 17.0 | 4 |
| | | | | 25 | 6 | | 5 | 4+8 | | | |
| | | | | 35 | 6 | | 10 | 4+8 | | | |

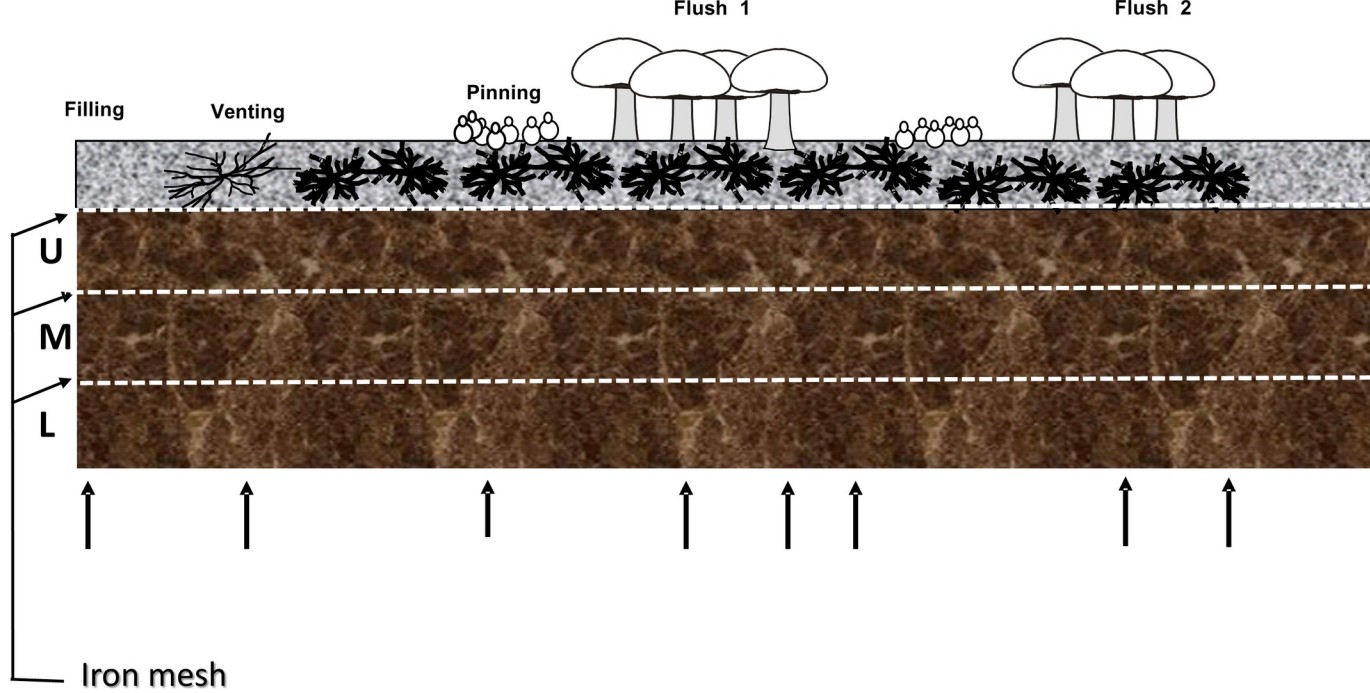

**Fig 1. Schematic diagram of a cultivation bed for button mushrooms.** The upper layer (grey) consists of casing soil that covers a layer of substrate (brown). At three different heights, iron grids were placed, one between the casing soil and the substrate and the two others at two different depths in the substrate dividing the substrate into three equal layers. On top a schematic visualization of the different crop phases and the upright arrows indicate the time points in the crop phase at which the substrate was sampled for analysis.

(fresh weight). The substrate of each layer was subsequently well mixed, and a sample dried at 105˚C until constant weight to determine moisture content and dry weight.

## Experiment 2: Effect of different substrate depths on yield and quality of mushrooms

Trays of 0.1 m$^2$ surface area were used filled with phase III substrate at three different levels 8.2, 6.7 or 5.2 kg substrate per tray (equivalent to 82, 67 and 52 kg substrate/m$^2$ and depth of approximately 17.7, 14.5 and 11.2 cm). For each substrate filling level, three levels of additional N-rich supplements (Substradd) were used to reach end concentrations, 20, 25 or 35 kg supplementation per tonne substrate. This increase in concentration was done to see in how far the extra supplementation can compensate the reduction in yield expected by reduction of substrate per square meter. Each substrate/supplement combination was done in duplicate resulting in 18 trays. Casing was done as described in the previous paragraph.

## Experiment 3: Interruption contact growing mushrooms with substrate

This experiment consisted of seven treatments (1–7), each in four-fold. Trays of 0.2 m$^2$ were filled with 17 kg phase III substrate and cased as described in 2.1. Treatment 1 is a control. Iron grids were placed either between the casing soil and the substrate (treatment 2 and 3) or at two depths in the substrate dividing the substrate in three equal layers (Treatments 4–7; Fig 2A). In treatment 2, 4, and 6, the grids were shortly lifted daily during five days from venting up to pin formation to disrupt contact between growing mushrooms and all or part of the substrate. In treatments 3, 5 and 7, the grids were shortly lifted daily during 10 days from venting

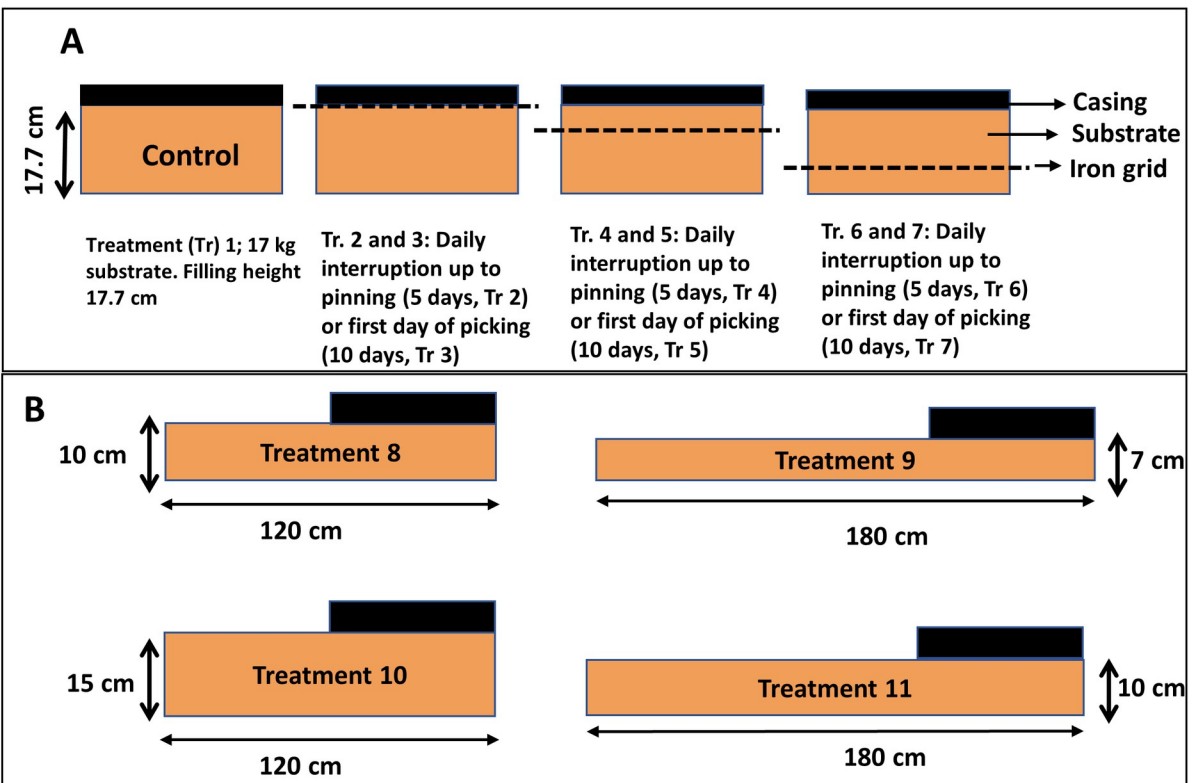

**Fig 2.** A: Trays used for a short daily interruption of the connection of the casing layer with all of the substrate or part of the substrate from venting up to the formations of pins (5 days, treatments 2, 4 and 6) or from venting up to the harvest of the first flush (10 days, treatments 3, 5 and 7). B: Trays with a sidewise extension of the substrate. The cultivation area (casing soil) is a fixed area and identical to those in A whereas the height of the substrate and length of the sidewise extension varies as indicated.

up to picking the first mushrooms. Two flushes were produced and mushrooms divided in quality classes as described in 2.1. The number of mushrooms was counted per flush for each class. To determine dry weights of mushrooms, five mushrooms per tray were sampled, sliced and incubated for 24 hours at 45°C, 24 hours at 70°C and 48 hours at 105°C. Average piece weights of mushrooms were determined by dividing the yield per flush per quality class by the number of mushrooms. In order to maintain moisture content of the casing soil optimal, the crop of Experiment 3 and 4 was watered at day 1 (filling), and day 2 and 3 with 2 L/m$^2$. Same watering was done after the first flush (day 16 and 17).

### Experiment 4: Cultivation on trays with extended substrate

This experiment consisted of four treatments (8–11), each in four-fold. Data from treatment 1 of Experiment 3 (paragraph 2.3) were used as a control since experiments of paragraph 2.4 were carried out with the same substrate and mushroom strain, in the same room and at the same time as experiments of paragraph 2.3. Treatment 8, 9, 10 and 11 consist of trays with the same width (0.36 cm) as trays in the previous experiments but with different extended lengths and different depths (Fig 2B). Trays of treatment 8 and 9 contained nearly the same weight of substrate (20.6 and 20.7 kg) as did trays of treatment 10 and 11 (31 and 30.9 kg). In each treatment the same area (0.2 m$^2$) is covered with casing layer and the extended non-cased substrate covered with microperforated plastic to avoid too much evaporation. Data on production were measured as described in the previous paragraph.

## Biochemical analyses

**Ergosterol measurements to assess fungal biomass.** Ergosterol is a sterol specific for fungi and used often to estimate fungal biomass in organic matter [19, 20]. Substrate samples of 100 gram were taken from Experiment 1 from the upper, middle and lower part of the substrate, one sample per tray per time point (at filling, venting, pin formation, just after flush one was harvested, and just after flush two was harvested). To make the number of samples manageable to handle, samples from the upper and middle layer were mixed generating one measurement. Samples were freeze dried and stored at room temperature in the dark. Ergosterol was measured according to Gessner [21]. In short: 200 mg sample was saponified in 3 ml 10% KOH methanol for 60 min at 80˚C (glass tubes with screw caps, shaken at 500 rpm). After cooling to room temperature, to each sample 20 µl (0.5 µg/µl) 7-dehydrocholesterol (Sigma) was added to determine extraction efficiency. One ml MilliQ water and 2 ml hexane was subsequently added and samples mixed by shaking for 10 min (Labotech). After 10 min centrifugation at 4000 rpm, the hexane phase was transferred to a new glass tube. This was repeated once and the hexane phases pooled. The hexane was subsequently evaporated using a Thermo-Fischer Speedvac (60 min, 30˚C, speed 25). The dried material was dissolved in 10 ml methanol and filtered (0.2 µm) and immediately analysed in a HPLC (Waters HPLC-PDA); A Phenomenex Luna 5 µm C18 column (250 x 4.6 mm) with a C18 guard column (Torrance, CA) was used for the separation]. Solvent used was 90% methanol and 10% (1:1) 2-propanol/hexane. Injection volume of 0.5 µl was used at a flow rate of 0.5 ml/min, runtime 20 min, oven at 35˚C. The 280 nm absorbance was used to quantify ergosterol and expressed as µg ergosterol/gram dry substrate. For each tray (2) per crop phase time, 11 ergosterol measurements were done to account for variation in measurements.

**Laccase assay.** In the vegetative phase, the laccase activity is high and during mushroom formation this activity decreases considerably [11, 22]. To see how far substrate contributes in the sidewise extended part of the substrate, laccase activities were measured. From each tray of each treatment in Experiment 4 (paragraph 2.4), a 50 gram sample of substrate was taken in the middle from the non-cased part of the substrate at the peak of the first flush. Samples were taken from non-cased part at a distance of 20, 50, 80 and 100 cm from the edge of the cased area. Laccase activities in substrate were measured according to [23]. In short: 100 ml of demineralized water was added to 50 gram of substrate in a 600 Duran beaker and shaken for 20 min at 4˚C at 140 rpm. The suspension was filtered over a cotton cloth and subsequently over a nylon mesh (1 mm mesh width). The filtrate was centrifuged at 10,000 rpm for 10 min and the supernatant transferred to 2 ml Eppendorf vials and stored at -20˚C. Laccase activity was determined by measuring the change in $A_{525}$ with Syringaldazine as substrate ($\varepsilon = 65,000/$M/cm). The reaction mixture consisted of 0.04 M citric acid, 0.13 M $Na_2HPO_4$, pH 6.0 and 0.075 mM Syringaldazine. To 1 ml of the reaction mixture, 25 µl extract was added and the change in $A_{525}$ was measured with a boiled extract sample as a control. The linear part of the slope was used as a measurement of the relative laccase activities. For each sample the average of four laccase measurements were used (measure replica's). All activities were subsequently expressed as percentage of the maximum activity measured.

## Results

### Experiment 1: Changes in substrate at different depths during two flushes

The lower part of the substrate, and to a lesser extent the upper layer, increases in fresh weight and moisture content between filling and venting (Fig 3A & 3B). A substantial amount of water given during this period on top of the casing soils ends up at the bottom layer of the

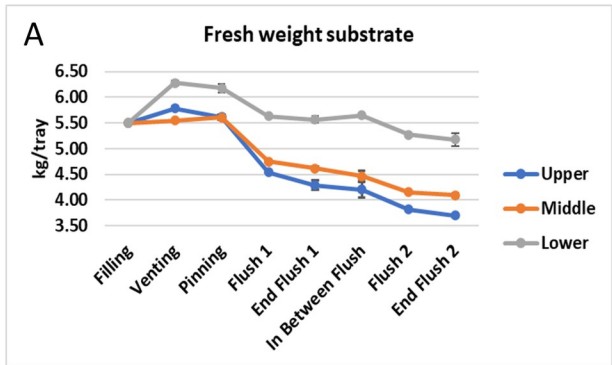

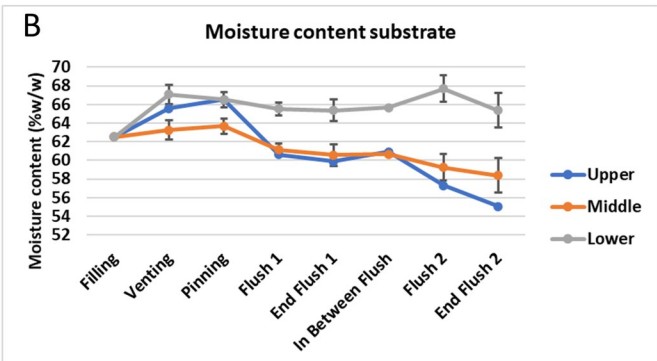

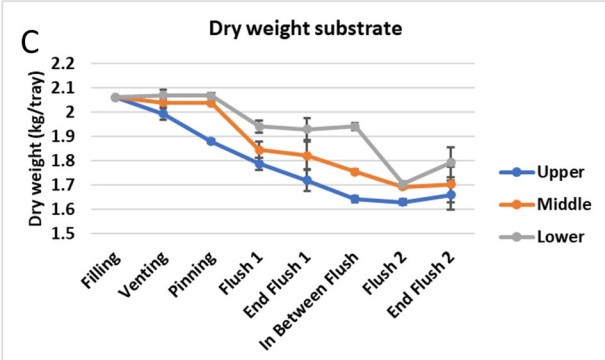

**Fig 3. (Experiment 1).** The changes in fresh weight (A), moisture content (B) and dry weight (C) of top, middle and lower layer of the substrate during the crop cycle of two flushes. Error bare represent ±1 SD.

substrate. In addition, water will be produced by the metabolic activity of the mycelium and some water will evaporate due to heat development. The upper layer remains wet up to pinning likely due to contact with the wet casing layer, whereafter the moisture level strongly decreases. During the development of flush one, fresh weight is decreasing in all layers but more in the upper and middle layer. That is for a large part due to water uptake by mushrooms from these layers. In the period from filling up to venting, only dry matter is degraded in the upper layer indicating that the colonization of the casing soil and pin formation is mainly fed by the upper substrate layer (Fig 3C). During the development of flush one, the degradation of dry matter continues in the upper layer at the same pace as the previous period, and now also degradation of the middle and the lower layer has started. During the production of flush two, the lower layer shows the largest decrease in dry matter and to a lesser extend the middle layer, whereas the upper layer does not show any further reduction in dry matter. The dry matter in the substrate is thus unevenly utilized during the production of the first two flushes. The upper and middle layer have the largest contribution in flush one while the lower layer has the largest contribution in flush two. The amount of dry matter degraded in two flushes in this experiment is 20 ± 3%, 17 ± 4% and 13 ± 4% in the upper, middle and lower layer, respectively.

## Experiment 2: Varying substrate depths

For each substrate depth (82, 67 and 52 kg/m$^2$), three levels of N-rich supplement (20, 25 and 35 kg/tonne substrate) was used. The level of supplementation had no effect on the total yield of mushrooms in three substrate depths in two flushes per square meter ($F_{(2,9)}$ = 2.546, p = 0.133), nor was there a significant interaction between substrate depth and level of

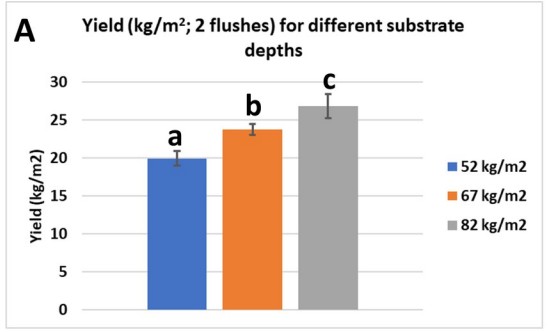

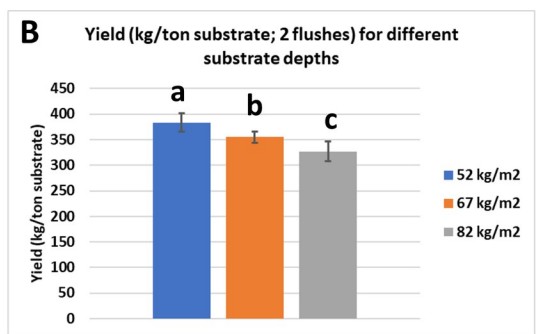

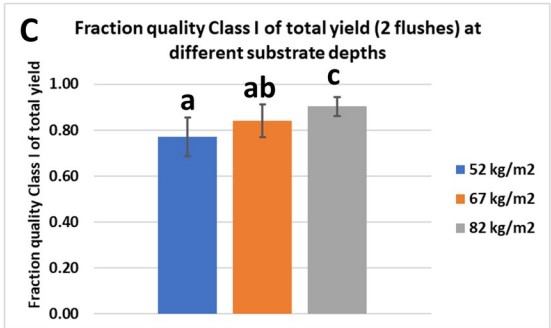

**Fig 4. (Experiment 2).** The effect of depth of substrate on the production of mushrooms in two flushes expressed as kg/m$^2$ (A), kg per tonne of substrate (B) and the effect of substrate depth on the quality of mushrooms (C). The quality is expressed as the percentage of the yield of Class I fine + Class I middle of the total yield (all classes). Data are means ± standard deviation. Significance at alpha = 0.05.

supplementation (F(4,9) = 1.078, p = 0.422, see S1 Appendix/Experiment 2 for statistical details). As expected, the differences in the total production in two flushes expressed as kg/m$^2$ were significant and decreased with decreasing substrate depth (Fig 4A). The production per tonne of substrate, however, increases significantly with decreasing substrate depth, Fig 4B). We see also a reduction in quality expressed as a fraction of Class I (fine & middle) mushrooms when filling less substrate (Fig 4C), although only the mushroom quality difference between substrate depths of 52 and 82 kg/m$^2$ was significant (p = 0.001). There is thus a positive correlation between substrate depth and yield/m$^2$ and quality on one hand and a negative correlation between substrate depth and yield/tonne substrate on the other hand (Pearson's correlations in S1 Appendix/Experiment 2).

## Experiment 3: Interruption of contact developing mushrooms with substrate

As an alternative to a permanent reduction of nutrients available for two flushes, we tested short interruptions of the nutrient flow towards pins and outgrowing mushrooms. That can be done since *A. bisporus* can quickly repair a mycelium network after disruption, such as is seen after filling shelves with phase III substrate. To carry out such experiments, we used again iron grids placed between the casing soil and the substrate or at two different depths in the substrate. Starting at the moment of venting, the grids were daily lifted for a short period either for five consecutive days (up to pinning) or for ten consecutive days (up to the first day of picking mushrooms; see Fig 2 in M&M). This breaks the mycelial network and the nutrient flow

towards the developing mushrooms is thus interrupted shortly, at different time points, from either all the substrate or from part of the substrate. We assume that these interruptions are short enough to not to interfere with the typical activity changes of lignocellulosic enzymes in the substrate during the mushroom formation, i.e., a decreasing laccase and increasing cellulase/hemicellulose activity [11, 22] and thus the total amount of nutrients available is not affected. However, the transport of nutrients to outgrowing mushrooms will depend on the rate of mycelial network restoration. The length of the period in which interruptions were done and the position of the iron grid had no significant influence on the total yield (kg/m$^2$) in two flushes (S3 Fig; See also S1 Appendix/Experiment 3). A daily disruption of contact between the casing soil and substrate during five days after venting shows a delay in production of the first flush compared to the control (Fig 5A) and an extension of the duration of flush two. Disruptions for ten days show a similar delay. The disruption of contact between casing and substrate also causes a shift in yield from flush one to flush two when disruptions were done for 10 days. When the disruptions were done with only part of the substrate, we do not see a delay in the production profiles (Fig 5B). Here we also see a shift of the production towards flush two correlated with the length of time during which the disruptions were done (for all treatments significant difference in yield of flush two between 5 days and 10 days disruptions ($p = 0.002$); no significant differences in yield of flush 2 between 0 and 5 days disruptions). For all treatments there seems to be a fixed amount of mushroom biomass produced in two flushes with shifts in yield from flush one to flush two due to disruption of contact between developing mushrooms with all or part of the substrate. As expected, this leads to a strong negative correlation between the yield in flush one and flush two (Pearson's Correlation $r = -0.847$; $p<0.001$; Fig 6A). There is a clear positive correlation between the number of mushrooms and yield for both flushes (Pearson's Correlation $r = 0.881$ for flush one and $r = 0.852$ for flush two; $p<0.001$; Fig 6B & 6C) indicating that the shifts in yields between flushes are mainly due to the number of mushrooms that are formed.

### Experiment 4: Sidewise extended substrate

In the fourth experiment the amount of substrate was varied without varying the cultivation area (substrate area covered with casing soil). The additional substrate was offered by extending the substrate sidewise (see Fig 2 in M&M and Table 2). In addition, the height of the substrate was varied. In this way it will be clear if the extra substrate offered is used for (and to what distance from) the developing mushrooms and if this extension can compensate for the reduced height of the substrate. The total yield in two flushes shows large differences between the treatments and has no correlation at all with the total amount of substrate offered but a clear increase in yield is seen with an increasing amount of substrate underneath the casing soil (Fig 7A). Yields expressed per tonne substrate, considering only the substrate underneath the casing soil, decreases with increasing substrate depths, as seen in Experiment 2, indicating a more efficient use of substrate with a lower depth of substrate (Fig 7B). In addition, here the quality also decreases with decreasing filling height of substrate although the qualities differ only significant between the two highest and the lowest substrate depth (Fig 7C). The lack of contribution of the sidewise substrate extension to yield is underpinned by the laccase activity in these extensions. In the vegetative phase, the laccase activity is high and during mushroom formation this activity decreases considerably [11, 22]. The laccase activities at the peak of flush one between treatments where not significantly different for the same distances from the casing soil, nor was there a significant interaction between treatment and distance to the casing soil (see S1 Appendix). Laccase activity was low in samples taken at 20 cm distance from the cased part and high in samples taken 50, 80 or 110 cm distance from the cased part (significant

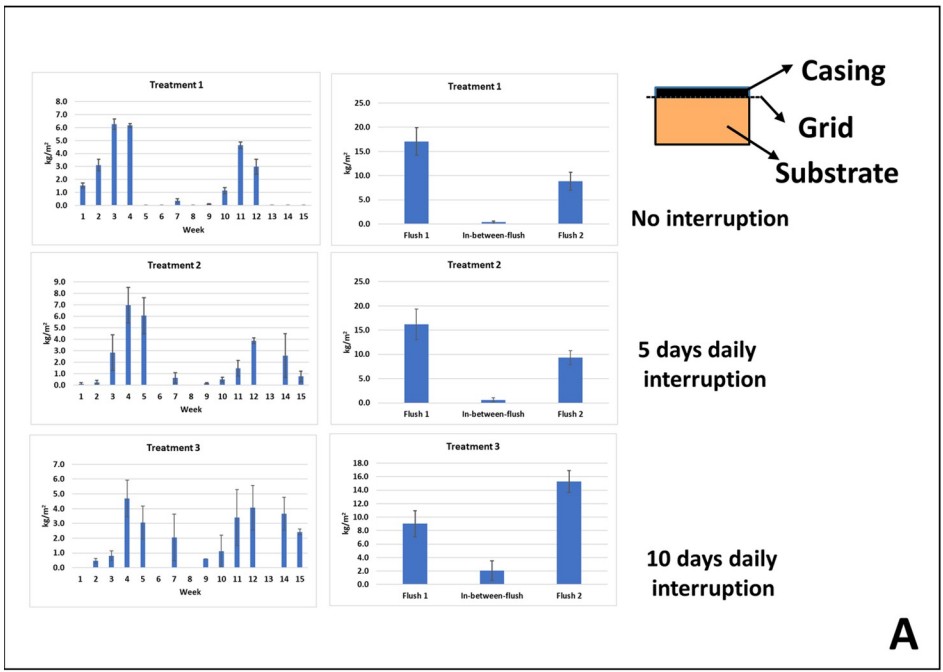

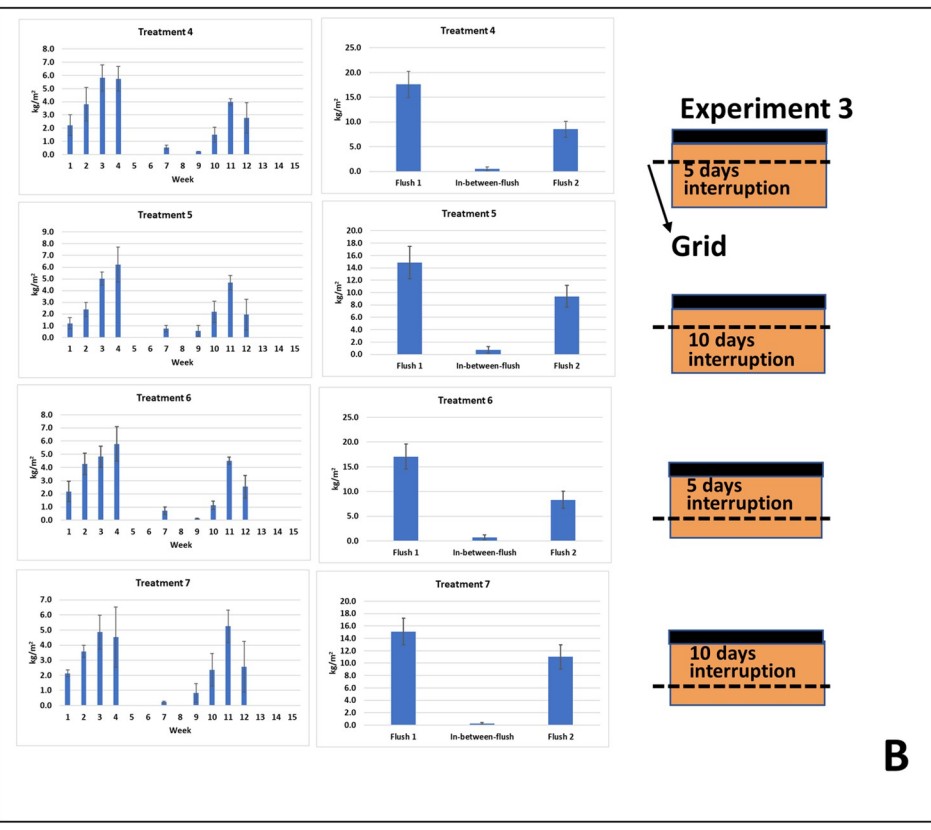

**Fig 5. (Experiment 3).** The production profile (left, kg/m$^2$) and the yield per flush and the in-between-flush (right) of the treatments where the connection of the casing soil with all of the substrate (A) or part of the substrate (B) where daily shortly interrupted. Compared to the control with no interruption (Treatment 1), a clear delay of the first picking day is seen when the connection of the casing soil with all of the substrate in interrupted (treatments 2 and 3). No delay in production of flush one is seen when interruptions were done with only part of the substrate (5B). All treatments

show a more or less shift from the yield in flush 1 towards flush 2 compared to the control (treatment 1). Error bars represent ± 1 SD.

difference between 20 cm and 50/80/110 cm: p<0.001; no significant differences between 50, 80 and 110: p = 0.9; Fig 8). This indicates that only at a short distance from the mushroom producing site the enzyme activity is in accordance with what is expected for mycelium connected to developing mushrooms.

## Mycelial biomass

The ergosterol content in the substrate of Experiment 1 was measured during the crop cycle of two flushes as an indication of mycelial biomass formation. Samples were taken from the upper, middle and lower part of the substrate. To generate a manageable number of samples for the laborious ergosterol analysis, we mixed samples from the upper and middle layer leading to two data for each timepoint. There were no statistical differences between the mixed upper/middle samples and the lower samples nor was there interaction between layer and phase in the crop (see S1 Appendix). During the time between filling phase III substrate and the onset of pin formation, we see an increase in ergosterol (Fig 9). At the time pins develop into mushrooms, we see a steeper increase which levels off to a fairly constant amount during flush two. The ergosterol concentrations are significantly different between filling and pinning (p = 0.009) and between pinning and after flush1/2 (p<0.001).

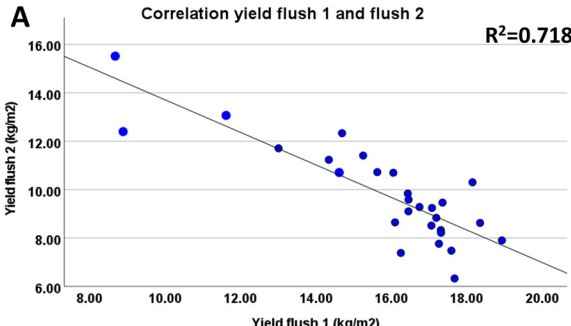

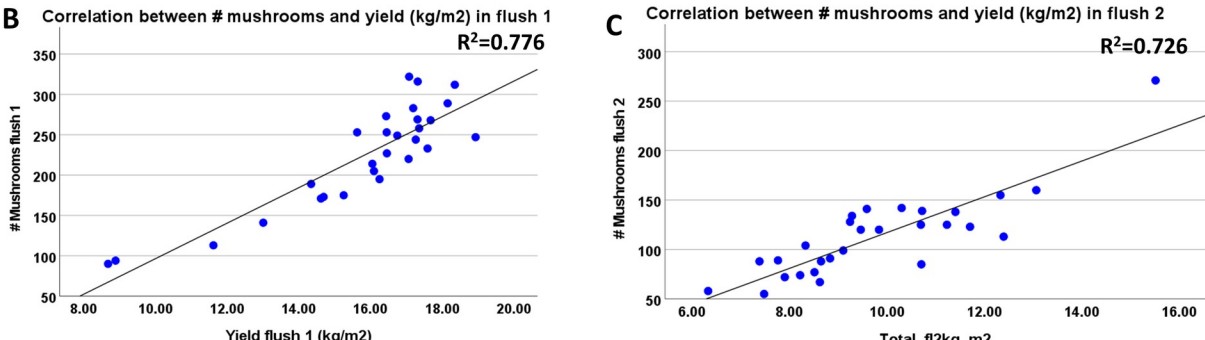

**Fig 6. (Experiment 3).** A: The short disruption of contact between the casing soil and all or part of the substrate causes shifts in yield from flush one to flush two. This leads to a strong negative correlation between yield in flush one and flush two (Pearson's Correlation r = -0.847). B&C: The effect of the interruption of the connection between the casing soil with all or part of the substrate on the yield is mainly caused by the change in the number of mushrooms formed in flush one and flush two. A strong correlation is seen between the number of mushrooms and the yield in flush one (Pearson's Correlation r = 0.881) and in flush two (Pearson's Correlation r = 0.852).

**Table 2. Substrate depth, substrate underneath the casing soil (kg/tray) and the total amount of substrate/tray) for all treatments.**

| Treatment | Substrate depth (cm) | Substrate underneath casing (kg) | Total substrate (kg/tray) |
|---|---|---|---|
| 1 | 17.7 | 17 | 17 |
| 8 | 10 | 10 | 20.6 |
| 9 | 7 | 7 | 20.7 |
| 10 | 15 | 14 | 31 |
| 11 | 10 | 10 | 30.9 |

## Discussion

The analysis of substrate degradation at different depths during a crop cycle of two flushes shows that the substrate layer is unevenly utilized. The reduction in dry matter shows that the upper and middle layer have the largest contribution to flush one while the lower layer has the largest contribution to flush two. Thus there is a gradual utilization from top to bottom with advancement of flushes. The changes of moisture content and dry matter in the substrate during the two flushes are not parallel indicating that the ratios of uptake of water and nutrients differ for different layers during the crop cycle and vary per time point. Kalberer [24] studied moisture content, water- and osmotic potentials in three layers of substrate during two flushes. The changes in moisture content in the three layers he measured were similar to what we have measured here. The water potential, however, showed in his analysis the largest decrease in the middle layer after two flushes. Kalberer concluded that this indicates that *A. bisporus* show the highest metabolic activity in the middle layer. We measured the degradation of dry matter in the three layers and showed a decrease in substrate degradation from top to the bottom layer in two flushes. That indicates that the measurement of water potential alone does not generate a good indication of substrate utilisation. Smith and colleagues [25] used a trough system with a substrate layer of 0.9 meter to study nutrient transport during mushroom production. They

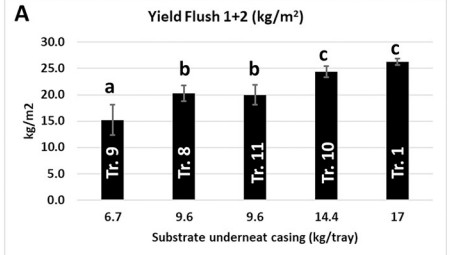

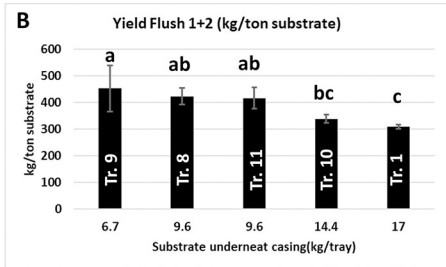

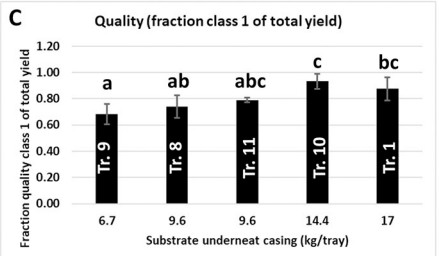

**Fig 7. The mushroom yield (flush one + two) in the experiments with a sidewise extension of the substrate.** The X-axes represent the amount of substrate underneath the casing soil. A: There is an increase of the yield in two flushes with an increase of the amount of substrate underneath the casing soil. B: There is also a tendency of increasing yield per tonne of substrate with a decreasing of the amount of substrate offered underneath the casing soil. C: As seen before, there is a tendency of a decrease in quality with a decrease in substrate depth (significant differences in fraction quality Class 1 between the two highest and lowest substrate depths (p≤0.01). Error bars represent ± 1 SD.

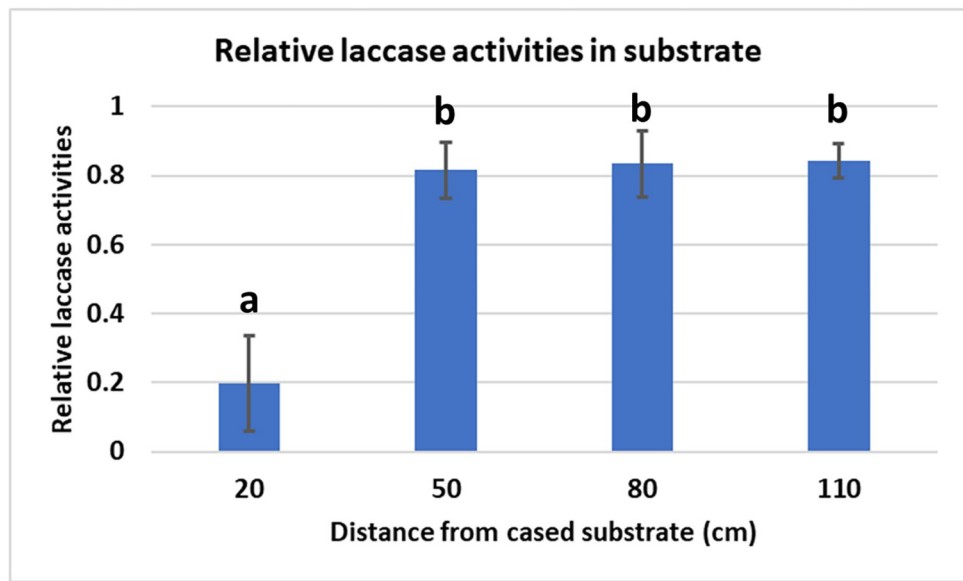

**Fig 8. Relative laccase activity in the sidewise substrate extensions measured at the peak of flush 1.** The high activities at a distance >20 cm away from the casing soil indicate that beyond this distance the substrate mycelium is not under the influence of fruiting and not involved in feeding mushrooms (the mean difference significant at 0.001 level). Error bars represent ± 1 SD.

reused substrate from different depths (five equal layers) after the second and the fourth flush in a new crop cycle to study the "residual crop potential". This showed a gradual increase in yield on layers taken from top to bottom indicating that the substrate is utilized from top to bottom with increasing flush number. Although less dramatic, we see such an effect thus even in a shallow layer of 17.5 cm during two flushes in our experiments.

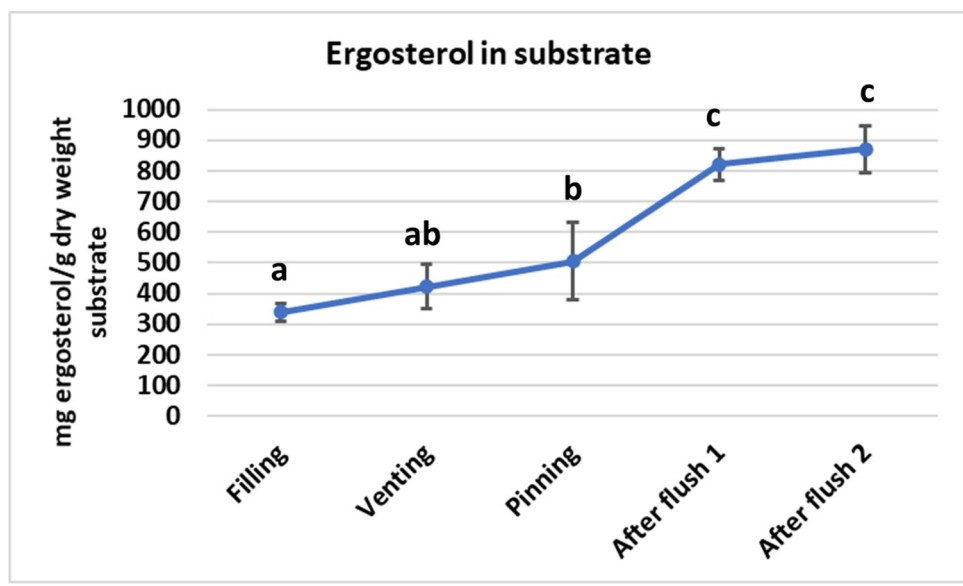

**Fig 9. The ergosterol content of the substrate during the crop cycle of two flushes (samples taken from Experiment 1).** Especially the steeper increase during the development of flush one indicates a mycelial growth (probably branching) during the development of this flush which levels of after flush one and remains approximate the same at least until after flush two (the mean difference significant at 0.05 level). Error bars represent ± 1 SD.

The short disruption of contact between the casing layer and all or part of the substrate (Experiment 3) had no influence on the total yield in two flushes but caused a shift in yield from flush one to flush two. Treatment 3 (ten days of disruptions) shows the most extreme shift of yield leading even to a higher yield in flush two than in flush one. This indicates that there is a fixed amount of nutrients freed by degradation of substrate and available for the production of mushrooms in two flushes. The applied disruptions apparently interfere with the nutrient transport towards the formation and/or outgrow of pins and mushrooms after the induction of fruiting. This results in a reduction of the number of pins that can develop into mushrooms in the first flush, and the nutrient surplus is utilised in the second flush. When the disruptions were done with only a part of the substrate, we do not see any delay in the production profiles (Fig 6B). This is consistent with the observation that the onset of fruiting (pinning) is fed by only the upper substrate layer. Since there is a clear correlation between the number of mushrooms and the yield in flush one and flush two, one would expect also a high correlation between the number of mushrooms in the two flushes. There is, however, no clear correlation between the number of mushrooms in flush one and flush two ($R^2$ = 0.256). That indicates that mushroom size might have an influence too. For flush one there is a clear negative correlation between piece weight and number of mushrooms for quality Class I fine and middle (r = - 0.648 and– 0.724, respectively; p<0.001; S1B and S1D Fig), whereas this correlation is absent for flush two (S1C and S1E Fig). For Class II mushrooms there is a moderate (flush one) and stronger (flush two) negative correlation between piece weight and number of mushrooms (r = -0.377 and– 0.523, respectively; S1F and S1G Fig). Especially the Class I mushrooms tend thus to be larger when less mushrooms are formed in flush one, a phenomenon also used by growers to produce large/giant quality mushrooms. They reduce the number of mushrooms by removing pins in an early stage which allows the remaining pins to develop into large mushrooms of high quality. The quality of mushrooms in flush one and flush two of each treatment was expressed as a percentage of Class I mushrooms (fine and middle) of the total yield (including Class II). There is some variation in mushroom quality between treatments, but these are not significant (p>0.05). As seen in previous experiments and also experienced by growers, the quality of flush two is clearly less than that of flush one (fraction Class I of all classes of flush one: 0.929; of flush two: 0.7287; paired sample t-test p<0.001; S2A Fig). Mushrooms in flush two also clearly have a lower dry weight than mushrooms in flush one (flush one: 7.43 (% w/w); flush two: 6.65 (% w/w); p<0.001; S2B Fig). There is no clear correlation between the number of mushrooms and their quality in the range of substrate depths used here. The major effect of the repetitive interruptions between casing and all of the substrate is thus a delay of flush one, a shift in yield from flush one to flush two during 10 days of interruptions mainly caused by a shift in number of mushrooms that are somewhat large in flush one.

The sidewise extension of the substrate (Experiment 4) was done in trays with a lower substrate depth than the control trays (Fig 2) in order to see how far these extensions could compensate for the reduction in the height of the substrate layer. The yield in kg/m$^2$ for two flushes corresponded highly with the substrate depth (kg/m$^2$) of the substrate layer underneath the casing soil (p<0.001) and not with the amount of substrate offered in the sideway extension. The laccase activity at the peak of flush one measured in the extended part of the substrate (treatment 8–11) showed a low activity at 20 cm and high activity at 50 cm or larger distances from the cased part of the substrate indicating that only a small part of the extended substrate was under the influence of the development of fruiting bodies. This does not proof *per se* that the extra 20$^+$ cm of the extended substrate also contributes to feeding mushrooms but does strongly indicate that the region in which the enzyme activities (gene expression) are regulated according to the crop phase is limited to this distance. Despite the "mushroom feeding" enzyme activity of 20 cm extended substrate region, this region seems not to contribute

significantly to the yield in the first two flushes. That makes sense since most of the substrate in the sideway extension measured from the edge of the casing soil is more than 20 cm away from most of the mushroom production site. When the sideway extended substrate would have been offered as increase of the filling heigh, thus placed directly underneath the production, all the additional substrate has a shorter distance to the production area and might have a positive effect on yield. One would expect that the small part of the substrate extension that shows the low laccase activity might contribute somewhat to the area close to the extension. We did not measure the yield distribution on the trays but observed that the mushrooms next to the extension have a better quality than mushrooms at the opposite site. The latter have a more stretched velum, indicative for a later developmental stage (S4 Fig). The higher quality of mushrooms adjacent to the extension indicates thus that the extra substrate of the first 20 cm contributes somewhat to feeding mushrooms.

## Mushroom quality and competition for nutrients

Straatsma et al. [17] showed that there is an exponential growth of mushroom biomass until this reaches 200 gram/kg substrate. After that point growth becomes linear indicating a competition for nutrients leading to maturation of mushrooms (opening of caps). A standing crop reaches a maximum of biomass of 400 gram mushrooms/kg substrate. There is thus a strong linkage between the amount of available nutrients (substrate depth) on one hand and growth and quality of mushroom biomass on the other hand. One would thus also expect a fairly constant ratio between the substrate depth and the yield and quality of mushrooms per tonne substrate. In experiment 2 and 4, where we applied various substrate depths, we see an increase in yield per tonne of substrate correlating with a decrease in substrate depth. The differences are not large but significant. We cannot exclude that this is, at least in part, due to the picking regime in these experiments. Mushroom pickers estimate quality mainly on size of mushrooms and less on maturation (i.e., stretched velum). If mushroom mature at a smaller size on lower substrate depth, mushroom should have been picked earlier, i.e., at a smaller size to ensure the same quality. That will reduce the total yield and improve the quality of mushrooms and might meet the expectation of a constant ratio between substrate depth and yield/quality of mushroom biomass. The remaining effect of a decreasing substrate depth is than the smaller size of quality mushrooms. The constant ratio between substrate depth and maximum yield can be underpinned by using standing crops for different substrate depths and measure the maximum amount of mushroom biomass formed.

## Gene expression and enzyme activities

Research on CAZyme gene expression in substrate during the crop cycle is very scarce. Patyshakuliyeva and colleagues [11] were the only, to our knowledge, that measured gene expression of the main CAZymes involved in substrate degradation during two flushes of the button mushroom. The expressed genes involved in (hemi)cellulose degradation (Fig 10A) show a relatively low expression from filling to pinning but a high expression during flush one. After flush one and during flush two, expressions are low again and increase only after flush two has been harvested. The RNA-seq data and qRT-PCR data correlated highly underlining the robustness of their data. They also carried out proteomic analysis of the same CAZymes at the various growth stages during two flushes. The abundance of the secreted enzymes remains high after the gene expression has decreased after flush one, indicating that the enzymes produced during flush one likely remains active after flush one and during flush two (Fig 10B). Filamentous fungi secrete proteins primarily at growing hyphal tips [26, 27]. The gene expression profile coincides to a large extent with our assessment of the formation of fungal biomass and is thus in accordance with the idea that mainly actively growing hyphae excrete enzymes. The

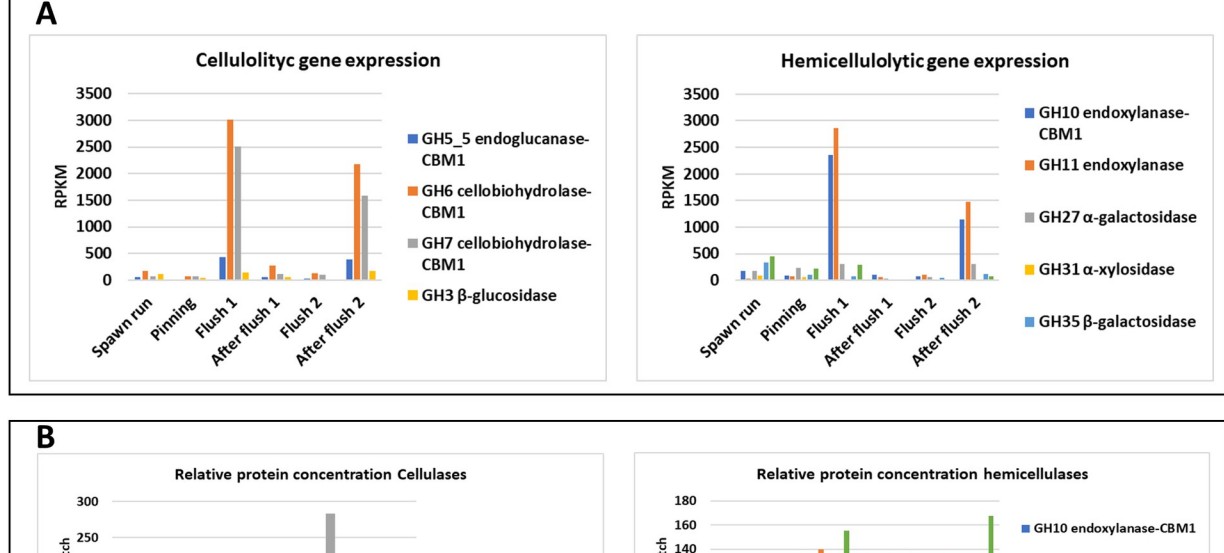

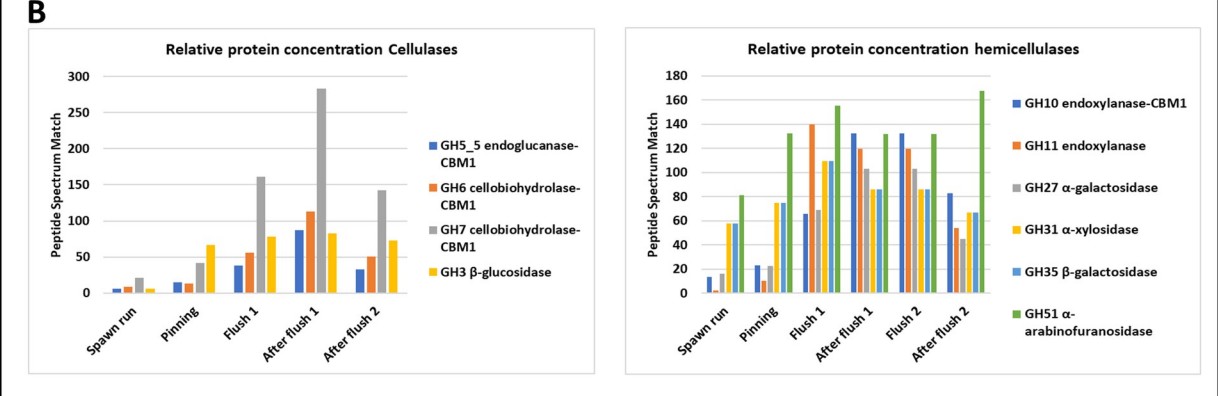

**Fig 10.** The expression of the major CAZyme genes (A) and their protein concentrations (B) during the crop cycle of two flushes after Pathyshakuliyeva et al. [11]. Expression of genes is expressed in RPKM (Fragments Per Kilobase Million). A clear peak is seen in the gene expressions during flush one which resumes only after flush two. This coincides with the formation of new mycelial biomass in the substrate. The protein concentrations, however, remain fairly high indicating that the enzymes, after secretion, remain active.

majority of enzymes needed to provide nutrients for both flushes must be formed before and during the development of flush one. The increase in mycelial biomass is likely realized by a strong branching of hyphae in order to create more tips for excreting enzymes. The increase in expression of CAZyme genes after flush two might indicate that for a third flush again new mycelial biomass is formed. To our knowledge, only Ohga et al. [28] studied also transcription regulation in substrate during fruit body formation. They assessed the mRNA levels of a laccase (*lcc2*) and a cellulase (*cel3*) gene in substrate mycelium during the crop cycle. The substrate was sampled at different stages of fruit body development in flush one. The laccase mRNA levels are high before venting (induction of fruiting) and steadily decrease after the onset of pinning and development of mushrooms into senescence in flush one. The cellulase (*cel3*) mRNA concentration shows the reverse pattern. This is in accordance with the experiments of Patyshakuliyeva et al. [11]. In contrast with the analysis for flush one, Ohga et al. (1999) analysed only one sample for laccase and cellulase mRNA expression for flush two and it is not clear at what time this sample was taken from the substrate. If it represents the middle of flush two, then the transcripts for cellulase mRNA levels are much lower than those shown for a button/veil-break stage mushrooms of flush one. That is similar to what Patyshakuliyeva et al. [11] have found. If the sample was taken at the start of flush two (pinning), then the mRNA levels are similar to that of the pinning stage of flush one and no conclusion can be made on the mRNA levels later in flush two.

We could only find three publications on extracellular cellulase activities in the substrate during the crop cycle [22, 29, 30]. Although the authors in all three papers state that there is a clear correlation between extracellular cellulase activities and fruit body biomass formation, this exact correlation is not always evident from the relevant graphs presented. Wood & Goodenough [29] conclude that "*the cellulase activity remains low until after the first pins were seen and then increases some 10-fold. The activity remains high for some time and then declines during the later cycles*". What is actually shown in the relevant graph is an increase of cellulase activity a few days after the first pins appear, with a steady increase in activity during ca. 10 days, a relatively high activity during the subsequent 10 days and then a sharp decline in activity within 2–4 days, a period that spans a time for two to three flushes in a regular crop. The authors only indicated the period for the first flush in the graph and not for the subsequent flushes. This first flush is marked in the graph as a 7–8 day period starting at the pin stage. This flush one covers thus only the first part of the steep increase in cellulase activity and the activity increases substantially after the first flush. The authors did not explain their harvesting strategy and the long period of increase in cellulase activity might indicate that the mushrooms were allowed to mature and even become senescent. If so, then the cellulase activity would correlate with the fruit body biomass formation.

Claydon et al. [22] did describe their harvesting strategies in more detail. In their first experiment, mushrooms were harvested as in a typical commercial crop. They state "*cellulase activity levels rose and fell in direct proportion to harvested fruit body mass*". What is actually seen in the relevant graph is a peak of cellulase activity during the first two flushes and cellulase activity decreases sharply only after flush two. The activities seem to follow the mushroom biomass production in the next two flushes. A short period of steady level of cellulase activity is seen between flush one and two and the activity increases again with the development of flush two. For the first two flushes there is thus not a good correlation between cellulase activity and fruit body mass production since there is no sharp decrease in cellulase activity between flush one and two. A better correlation was seen when Claydon and colleagues used a different harvesting strategy. Harvest of mushrooms of the second flush were delayed for one week leading to a dense crop of senescent mushrooms. This led to a high cellulase activity, 2-3-fold higher than in flush one. After removing flush two the cellulase activity decreased sharply. This experiment and that of Wood & Goodenough have in common that when mushrooms of a flush are allowed to mature to senescence, the cellulase activity correlates indeed with the formation of biomass. Since such senescent flush represents a much higher biomass it is conceivable that all the nutrients available are consumed by this large flush. In a crop where quality mushrooms are picked, each representing less biomass, the nutrients are distributed over these two flushes. The correlations between cellulase activities and flushing do not correlate with the proteomic data from Patyshakuliyeva et al. [11] and might be explained by inactivation of CAZymes.

Smith and colleagues [30] measured enzyme activity in a deep trough system (0.9 m) during five flushes. The data from the upper 30 cm (comparable to experiments of Claydon et al. and Wood & Goodenough) show a peak of cellulase activity in flush one, three and five and a dip or low activity during flush two and four. Although their data do not show the amount of fruit body biomass in each flush, there seems thus also not a complete correlation between cellulase activity and fruit body mass production. Although the variation in cellulase activity during flushes is not always identical in the three articles, it is clear that the cellulase activity is induced after the onset of mushroom formation but the synchrony in cellulase activities and mushroom biomass formation is not always clear.

Smith et al. [30] showed in their trough system that there is an increase in endocellulase activities in the deeper layers (50 and 70 cm deep) in later flushes, but these do not reach the levels seen in the two upper layers and might explain why the mushroom yield gradually

decreases with increasing number of flushes. They also measured the laccase activities at five different depths during five flushes. The laccase activities in the upper layers show an opposite pattern of the endocellulase activity. Laccase activity is high before flush one and fluctuates at a lower level during flushes with lower activity at a peak of a flush and a higher activity between flushes. The decline in laccase activity at the onset of fruiting and the low activity at flush peaks is only clear in the three upper layers. At a distance more than 50 cm from the fruiting area the laccase activity remains high. In our experiments with a sidewise extended substrate we see a high laccase activity at the peak of flush one somewhere between 20 and 50 cm away from the mushroom production area which correlates with their observation on depth.

Herman et al. [14] used a metabolically inactive labelled amino acid ($^{14}$C-AIB) to study transport in *A. bisporus*. They demonstrated that linear (directional) growth shows a faster transport and over a longer distance than non-directional (radial) growth of mycelium. In directional growth (substrate inoculated at one side) more mycelial cords/strands are formed that have a higher transport capacity. The maximum detectable translocation distance of $^{14}$C-AIB was between 50 and 99 cm in directional and 22–49 cm in non-directional growth. The latter type of growth represents phase IV substrate (mushroom producing substrate) that feeds mushrooms and agrees with our observations of a laccase activity that is low at a distance of > 20 cm away from growing mushrooms and beyond this distance the substrate does not contribute to the feeding of mushrooms in a crop of two flushes.

The number of primordia formed at an early stage is a manyfold higher than the number of mushrooms harvested in two flushes. It is likely that the number of primordia is sufficient for several flushes and there are indications that new pin formation is not needed for subsequent flushes [17, 31] although it is not proven that primordia formation stops during the outgrowth of the first flush. Straatsma et al. [17] observed that "*not all primordia that start to grow out reach the mature stage and quite a number is arrested. Arrest occurs before or at a size of about 10 mm diameter. Arrest indicates nutritional competition.*" The interruption we applied from venting to pinning or to the first day of harvest clearly interferes with the number of pins that develop into mushrooms in the first flush and can thus be explained by assuming the interruptions reduce the available nutrients leading to a reduced number of pins that can develop. The interruptions do not cause a difference in total yield in two flushes but only a shift in yield from flush one to flush two and this shift is almost completely due to the difference in number of pins that develop into mushrooms. This "yield compensating" relationship between flush one and two indicates that a fixed amount of nutrients is available for the first two flushes and this might be related to the fixed amount of mycelium that is formed corresponding with a fixed amount of excreted CAZymes at the start and during the development of flush one. If in flush one the number of pins that develop into mushrooms is restricted due to an interrupted supply of nutrients, the surplus will be used for the outgrow of extra pins into mushrooms in the next flush. The reverse is also observed by Straatsma et al. [17]: when mushrooms are picked at a very early stage (8 mm diameter), the number that is picked in the first flush is approximately twice the number of mushrooms picked with 40 mm diameter. The removal of small outgrowing primordia reduces competition for nutrients. Once primordia reach the size of 10 mm, however, they grow exponentially indicating no competition for nutrients. The number of pins that reach this size is thus regulated to allow an exponential growth.

## The limits of the present system

The production system with its present substrate has clear limits. The optimal substrate filling weight (substrate depth) has a quite narrow range, somewhere between 85 and 110 kg/m$^2$. Less reduces yield/m$^2$ and thus reduces efficiency of space use. More substrate leads to higher costs

without a higher return in crop value. In addition, temperature control will be difficult in too high substrate layers. The indoor fermentation of raw materials in the two first phases lead to a loss of almost 21% of dry matter and during the colonization of phase II substrate by *A. bisporus* an additional 9% of dry matter is lost [1]. During the production of two flushes a further 17% dry weight is degraded. As a consequence, only 11% of the dry matter of the original source materials is used to produce mushrooms, a meagre biological efficiency.

Herman et al [14] estimated the network transport capacity considering the water pressure difference between substrate and expanding mushrooms and the structure and number of cords (the main transport channels) in the mycelial network. They suggest that the transport network in the present system has reached its limit for the number mushrooms produced in such a short time course. Looking at flush one in a typical commercial crop, the immediate impression is also that there is hardly room for more mushrooms on a growing bed. An improvement of the efficiency of the production system can only come from a more nutrient rich substrate in which the conversion per unit of substrate into mushrooms is much higher and thus less substrate is needed per unit mushroom produced.

Heaton et al. [32] developed a model, using energy spend on (vegetative) growth, reproduction and substrate digestion, to identify the strategy to maximize the fraction of energy that could possibly be spend on reproduction (in our case fruiting). They show that on nutrient-rich media (agar plates) the total energy available for reproduction is much higher than for growth on nutrient poor (wood) substrates. In the presently used substrate, the button mushroom has to invest considerably in the synthesis and excretion of a number of enzymes to digest the substrate. According to the expression studies by Patyshakuliyeva et al. [11] at least 16 exoenzymes are expressed to degrade lignocellulose. Bechara and colleagues [33] experimented with the production of button mushrooms on several grain types. They optimized the colonization time of grains by *A. bisporus* by pre-incubating grains for 4–6 days with the thermophilic fungus *Mycothermus thermophylum*, a fungus that is known to promote vegetative growth of *A. bisporus* [7]. They reached biological efficiencies (BE: fresh weight produce/dry weight substrate) between 190 and 250% in one flush. For the conventional substrate BE ranges from 60–70% for one flush. A production of ca. 18 kg mushrooms/$m^2$, the former needs 21–28 kg substrate/$m^2$ and the latter 85–110 kg substrate/$m^2$. The obvious reason for this higher substrate conversion for grains is its nutrient composition and nutrient density. The major carbon source of grains is starch and the energy needed to produce enzymes for its degradation (amylases) is much lower than the energy needed to degrade lignocellulose (CAZymes).

The problem with nutritional dense substrate is obviously costs and absence of selectivity leading easily to infections. It would, nevertheless, be useful to do more research on alternative substrates that might considerably reduce the amount substrate needed to produce mushrooms.

## Conclusion

Our experiments indicate that in a harvesting strategy towards quality mushrooms there is a fixed amount of nutrients prepared/available to feed two flushes. The increase of mycelial biomass after venting and especially during the development of flush one indicates that likely a branching of mycelium occurs during this period, leading to a substantial number of growing hyphal tips excreting lignocellulolytic enzymes that degrade substrate and make nutrients available during flush one which are utilised in flush one and two. This is supported by the fact that the mycelial biomass increase coincides with a high gene expression for CAZymes [11], confirming that mycelium excretes enzymes mainly at growing tips. Previous experiments in

which cellulase activities were measured during flushing indicate that when a flush is allowed to fully mature (senescent mushrooms) the additional mushroom biomass formed in one flush consumes all of the available nutrients and likely new mycelial biomass must be formed to generate nutrients for additional flushes. This indicates that the amount of fungal biomass formed in the substrate is one of the determining and limiting factors for nutrient supply to growing mushrooms. Further studies on the assessment of fungal biomass under various harvesting strategies and during multiple flushes is needed to underpin the exact function of the mycelial biomass.

The presently used substrate seems to have reached its limits and there are hardly opportunities for further improvement. Research on alternative, more energy dense, substrates are needed to improve the efficiency of the present production system.

## Supporting information

**S1 Fig. Experiment 3.** A&C: A negative correlation in flush one between piece weight and number of mushrooms for quality Class I fine and middle (Pearson's Correlation r = - 0.648 and– 0.724, respectively; p<0.001), while such correlation is absent in flush two (B&D). E&F: For Class II mushrooms there is a moderate (flush one, E) and stronger (flush two, F) negative correlation between piece weight and number of mushrooms (Pearson's Correlation r = -0.300 and– 0.523, respectively).
(TIF)

**S2 Fig. Experiment 3.** A: Significant difference in quality of mushrooms in flush one and flush two (p<0.001). B: Significant difference in dry weigh of mushrooms in flush one and flush two (p<0.001).
(TIF)

**S3 Fig. The total yield (flush one + flush two) of the different treatments of Experiment 3.** The short daily interruptions of contact between the casing soil and all or part of the substrate did not lead significant differences in te yield after two flushes. The error bars represent 2 x the standard deviation.
(TIF)

**S4 Fig. Difference in quality of mushrooms grown near the extended part of the substrate and mushrooms grown at the opposite site.** The bottom of the figure shows a schematic diagram of a cultivation tray with a sidewise extension of the substrate. The lower photograph shows the mushroom bed at the time of picking of flush one. The photographic enlargements of the top left and right show that mushrooms close to the substrate extension (left) have a better quality (less stretched velum) than the mushrooms at the opposite site (right).
(TIF)

**S1 Appendix. Detailed statistical analysis.**
(XLSX)

**S1 Data.**
(DOCX)

## Acknowledgments

This project was sponsored by Productschap voor de Tuinbouw. We wish to thank Laura Berns for the analyses of ergosterol and laccase activities.

## Author Contributions

**Conceptualization:** Anton S. M. Sonnenberg, Johan J. P. Baars, Chris Blok.

**Data curation:** Anton S. M. Sonnenberg, Johan J. P. Baars, Patrick M. Hendrickx.

**Funding acquisition:** Anton S. M. Sonnenberg.

**Investigation:** Ed Hendrix.

**Supervision:** Anton S. M. Sonnenberg, Johan J. P. Baars, Patrick M. Hendrickx, Chris Blok.

**Writing – original draft:** Anton S. M. Sonnenberg.

**Writing – review & editing:** Johan J. P. Baars, Gerben Straatsma, Chris Blok, Arend van Peer.

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
