## [Decision Letter · Decision Letter 0]

18 Apr 2022

PONE-D-22-06925Feeding growing button mushrooms. The role of substrate mycelium to feed the two first flushes.PLOS ONE

Dear Dr. Sonnenberg,

Thank you for submitting your manuscript to PLOS ONE. After careful consideration, we feel that it has merit but does not fully meet PLOS ONE’s publication criteria as it currently stands. Therefore, we invite you to submit a revised version of the manuscript that addresses the points raised during the review process.

We look forward to receiving your revised manuscript.

Kind regards,

Anupam Barh, Ph.D

Academic Editor

PLOS ONE

Journal Requirements:

Additional Editor Comments:

The manuscript is technically sound and will be suitable for readers working on mushrooms. The manuscript title may be changed for better understanding for readers. Moreover a spell check and english correction is required for entire manuscript.

Reviewers' comments:

Reviewer's Responses to Questions

**Comments to the Author**

1. Is the manuscript technically sound, and do the data support the conclusions?

Reviewer #1: Yes

Reviewer #2: Yes

Reviewer #3: Yes

2. Has the statistical analysis been performed appropriately and rigorously? 

Reviewer #1: Yes

Reviewer #2: Yes

Reviewer #3: Yes

3. Have the authors made all data underlying the findings in their manuscript fully available?

Reviewer #1: Yes

Reviewer #2: Yes

Reviewer #3: Yes

4. Is the manuscript presented in an intelligible fashion and written in standard English?

Reviewer #1: Yes

Reviewer #2: Yes

Reviewer #3: Yes

5. Review Comments to the Author

Reviewer #1: The authors have planned the experiment very aptly to answer the questions on substrate degradation effect on the quality and quantity of the mushrooms. This research results have implications for both academicians and commercial mushroom growers. The authors have tried to support the study results with suitable literature review.

Reviewer #2: This is an interesting study and the manuscript was well drafted. Following are the few comments from my end

1. In view of the results mentioned in experiment -4 (from 297-299 lines), what should be the ideal depth of the substrate for the commercial production of Agaricus bisporus mushroom with the existing compost formulations?

2. It will be useful to the other co-workers, if the authors mention the compost and casing formulations used in the present study.

Reviewer #3: The manuscript is technically sound and adequate data using different field and lab based approaches has been generated, analysed and properly inferred.

The paper essentially deals with different facets of nutrient mobilization in button mushroom from different layers of substrate from casing to end of second flush. The title of the paper "Feeding growing button mushrooms. The role of substrate mycelium to feed the two first flushes." needs little correction and it can be Feeding growing button mushrooms: The role of substrate mycelium to feed the first two flushes.

There are minor corrections in the text as detailed below. At places there are unfilled dots to indicate that data is required here. These spaces may please be filled with appropriate data. Comments are mainly explanation for the comment and may be used for incorporating corrections wherever found relevant.

Line # : 2. Replace: two first flushes

With: first two flushes

Comments: The study deals with nutrient uptake in button mushroom from different layers from casing up to flush 2. The title can be more explicit.

Line #: 17-18

Replace:1/3

With:1/3rd

Line #: 28

Replace:20-50

With:20

Comments: In the absence of data at 30 and 40 cm, it is difficult to specify the exact distance

Line #: 46

Replace: matter by 16%, equivalent to 22% of the organic matter (2).

With: matter by 16%, equivalent to 22% of the organic matter (2) when compost is prepared using wheat straw ..%, chicken manure….%, ……gypsum…%.

Comments: This value of 22% is a function of the amount of gypsum used and mineral content of the straw or other substrate used and may be valid when ash content is around 27-28%. In many parts of the world paddy straw and other substrates are used and the quantity of gypsum also varies. Hence it is apt to give the exact formulation.

Line #: 62

Replace: In order to further our understanding…

With: To further our understanding…

Comments: Use of prefixes like In order to, in case of in stead of just To or In is a personal choice and is not needed at may places

Line #: 94

Replace: used.

With: used. The bulk density of the compost after Phase II and its N content was…… and ……

Comments: Bulk density of compost, its nutrient status and watering schedule can impact the nutrient uptake/utilization from different layers. Hence conclusions may be only broadly applicable to different compost formulations. It is thus apt to specify the parameters of the compost used and watering schedule from casing to end of second flush. It may be added at appropriate places.

Line #: 94

Replace: kg/ton

With: kg/ tonne

Comments: This may me corrected as per the pattern of Journal. At times ton and tonne are slightly different. Ton = 2240 pound whereas tonne is 1000 kg. (1000 kg =2204.6 pound)

Correction if agreed may be made throughout the paper at other places as well

Line #: below 97

Replace: 12

With: 4+8

Comments: In Table 1 Experiment 3 we have shown N as 12 where as it appears that we have 4 trays with no mesh and 8 trays with mesh between casing and compost (4 to be disturbed for 5 days and other 4 to be disturbed for 10 days). It is better if we write 4+8 at level 0 instead of 12.

Line #: 103

Replace: Trays with a surface area of 0.2 m2 were….

With: Trays with a surface area of 0.2 m2 (inside dimensions …..cm x ……cm) were….

Comments: We may give length x breadth. It is more relevant in Expt 3

Line #: 121

Replace: (equivalent to 82, 67 and 52 kg substrate/m2).

With: (equivalent to 82, 67 and 52 kg substrate/m2 and depth of …, …, and …. cm).

Comments: It will be apt if height is also mentioned. Or bulk density of compost is mentioned at any appropriate place

Line #: 146

Replace: same width as trays

With: same width (…..cm) as trays

Comments: Pl specify width

Line #: 147

Replace: but with different extended lengths and different depths

With: but with different extended lengths and different depths (……………….).

Comments: Pl specify the length at different depths

Line #: 184-185

Replace: Samples were taken at 20, 50, 80 and 110 cm from the edge of the non-cased part.

With: Samples were taken from non-cased part at a distance of 20, 50, 80 and 100 cm from the edge of the cased area

Comments: Though it is understood what is meant, but non-cased part has three free edges. So it is better to correct the sentence. In abstract it has been put properly

Line #: 186

Replace: 600 Erlenmeyer beaker

With: 600 Erlenmeyer flask

Comments: Normally Erlenmeyer is credited with flask and not beaker.

Line #: 188

Replace: 10.000

With: 10,000

Comments: I think common problem when we use different languages.

Line #: 217

Replace: is 20% ± 3,

With: is 20±3% or 20%±3 or 20%±3%

Comments: Please use which ever one is correct as per your data

Line #: 224

Replace: mushrooms per substrate depth in two flushes

With: mushrooms in three substrate depths in two flushes

Line #: 226

Replace: p=0.422,, see

With: p=0.422, see

Comments: delete ,

Line #: 229

Replace: We see also a in reduction of quality

With: We see also a reduction in quality

Line #: 257

Replace: When looking at the Yields expressed per ton substrate, considering only the

With: ??

Comments: Incomplete line. Can be deleted or merged with the previous one

Line #: 264

Replace: significant difference in yield flush two between

With: significant difference in yield of flush two between

Line #: 265

Replace: no significant differences yield flush 2 between 0 and 5

With: no significant differences in yield of flush 2 between 0 and 5

Line #: 344

Replace: contribution to flush two.

With: contribution to flush two. Thus there is gradual utilization from top layers to lower layers with advancement of flushes

Comments: Had we divided substrate into two layers only, then result possibly could have been:

During first flush top half layer is used, during period in-between two flushes nutrient from both layers are used and during 2nd flush material from the lower layer is mainly used. As we have tested only one system of 3 layers and not compared it with 2 or 4 layered system, hence, the conclusion needs to be balanced

Line #: 348

Replace: content in the tree layers

With: content in the three layers

Line #: 372

Replace: between te number

With: between the number

Line #: 389

Replace: ). There is no clear correlation between the number of mushrooms and their quality.

With: ). There is no clear correlation between the number of mushrooms and their quality up to yield level of ….ton/m2.

Comments: Overgeneralization. This is valid only at lower ranges. When the number of mushroom may increase beyond a level, these may affect quality

About Figures

Comments made are only suggestions and the ones deemed fit may be considered.

Fig 1 is a diagrammatic representation of different stages. To be more realistic we can show number of pinheads more than the mushrooms and number of mushrooms more in flush one than in flush two. However, pl see that it does not disturb the sampling arrows at the base.

Fig 2. Experiment 1. Premise seems to be that inserting mesh has no effect on fruiting. It would have been better to have one control for each mesh position to verify this premise. Alternatively, all the 28 bags could have three meshes at levels shown in Fig. 1. In that case, Tr 1 (Control): No mesh disturbed in four bags, Tr 2: top mesh disturbed for 5 days in four bags, Tr 3: top mesh disturbed for 10 days in four bags, Tr 4: middle mesh disturbed for 5 days in 4 bags, Tr 5: ….so on.

Please specify mesh/inch, wire dia, percent open area of mesh or other relevant specifications of the mesh used. (% open area = [1-(MxD)]^2 x 100 where M is mesh count or no. of wires per inch, and

D is diameter of wire in inches)

Fig 3 Moisture content is same in all three layers in the beginning as Phase III compost is used which gets homogenized at the time of filling. Do we expect the same scenario of phase II compost is used in trays as is the trend in many countries?

Pl specify what error bars represent (±1 SD)

In Fig 3C, increase in dry weight in lower layer at end of flush 2 needs some explanation, or it is just a non significant deviation.

Here high moisture content in the top and bottom layer and low in middle layer at pinning in Fig 3 B seems to be the outcome of way we water the crop. It will be apt if we can add information about water sprayed/ day/m2 and how many times/day on different days from casing till end of flush 2.

Fig 4A. Title needs to be corrected to Yield (kg/m2; 2 flushes… .

Apparently increase in yield/m2 is valid only up to a limit after which it may plateau or even start decreasing due to excess heat or carbon dioxide production. Any data from other publications if available may be included in discussion.

In Fig 4B yield kg/ton of substrate is an important aspect and the question commonly debated by growers that how much compost can be filled. This obviously depends on the cost benefit ratio as where the profits are higher, then even slightly less increase will be useful to the grower to get more mushrooms and hence more income per growing room. At the same time filling less compost not only decreases yield per room, but also affects the quality as discussed in the paper. Thus, it may be possible to define only the minimum filling depth. It will be apt if the same is made part of the conclusion and discussion.

Fig 5. It may be better to specify value at top of each bar (may be in vertical alignment mode). Absence of error bars suggests that it is sum total of all replicates. This will help in better understanding of the conclusion mentioned in the text that there is only shift in yield from flush 1 to 2 but overall yield of 2 flushes remains more or less same.

Fig 6B Secondary axis may be corrected as: # of mushroom in flush 1/m2

Fig 6C Primary axis may be corrected as : Yield Flush 2 (kg/m2)

Secondary axis may be corrected as: # of mushroom in flush 2/m2

Fig 9 has been titled as experiment 1. Instead of this in caption we can write that samples taken from Experiment 1

Fig 10: Secondary axis: RPKM can be expanded/mentioned in text

Fig 11: Looking at the dispersion of points, it will be apt to give R2 values for each graph

S2 Correct sec axis …………. Class (instead of Clas)

S2B Correct secondary axis as …Dry weight (instead of Fry …)

S4: Experiment 2: Normally SD values below a sample size are of less relevance and at the max indicate only degree of dispersion in data. In this experiment SD of 2 values has been calculated. Further, output up to 3 decimal is more than enough.

Total SD for each substrate kg/m2 seems to have been cultivated by just adding the basic data of all the three groups of supplementation. To compute the pooled SD from several groups, we normally calculate the difference between each value and its group mean, square those differences, add them all up (for all groups), and divide by the number of df, which equals the total sample size minus the number of groups. Its square root is the pooled SD. For example, for lines 19-21 we can pool SD =0.914011 (instead of 0.94073). Similarly pooled SD for 67 kg is 0.78 instead of 0.73 and for 82 kg it is 1.34 instead of 1.58. The differences are however miniscule and the values calculated have been used to only indicate the dispersion using error bars in bar or line diagrams. Due to low N (sample size), the df in different sources is low and it normally results in reduced efficiency of the test. In the present study, which is indoor study, it seems fairly OK to use less number of replicates considering the limited space available for experiments and uniformity of environment and other parameters during experimentation. This may be specified at any suitable place that the environment where experiments were cultivated were highly uniform and hence data from the limited trays was considered adequate.

6. PLOS authors have the option to publish the peer review history of their article (what does this mean?). If published, this will include your full peer review and any attached files.

Reviewer #1: **Yes: **Mahantesh Shirur

Reviewer #2: **Yes: **Dr Sudheer Kumar Annepu

Reviewer #3: **Yes: **Manjit Singh

---

## [Author Response · Author response to Decision Letter 0]

20 May 2022

We thank all reviewers for their comments and specially reviewer #3 for his thorough reviewing which has improved the mansucript.

---

## [Editor Report · Decision Letter 1]

15 Jun 2022

Feeding growing button mushrooms: The role of substrate mycelium to feed the first two flushes.

PONE-D-22-06925R1

Dear Dr. Sonnenberg,

We’re pleased to inform you that your manuscript has been judged scientifically suitable for publication and will be formally accepted for publication once it meets all outstanding technical requirements.

Kind regards,

Anupam Barh, Ph.D

Academic Editor

PLOS ONE

Additional Editor Comments (optional):

The corrections asked were made by authors.

Kindly check line number 492 "Ogha et al." change it to Ogha et al. (1999).

In line number 553 and 557 Please change "CAIB to C-AIB"

Best wishes

---

## [Editor Report · Acceptance letter]

30 Jun 2022

PONE-D-22-06925R1 

Feeding growing button mushrooms: The role of substrate mycelium to feed the first two flushes. 

Dear Dr. Sonnenberg:

I'm pleased to inform you that your manuscript has been deemed suitable for publication in PLOS ONE. Congratulations! Your manuscript is now with our production department. 

Kind regards, 

on behalf of

Dr. Anupam Barh 

Academic Editor

PLOS ONE